# Platelets Are Critical Key Players in Sepsis

**DOI:** 10.3390/ijms20143494

**Published:** 2019-07-16

**Authors:** Fanny Vardon-Bounes, Stéphanie Ruiz, Marie-Pierre Gratacap, Cédric Garcia, Bernard Payrastre, Vincent Minville

**Affiliations:** 1Anesthesiology and Critical Care Unit, Toulouse University Hospital, 31059 Toulouse, France; 2INSERM I2MC (Institut des Maladies Cardiovasculaires et Métaboliques) UMR 1048, Toulouse University Hospital, 31059 Toulouse, France; 3Hematology Laboratory, Toulouse University Hospital, 31059 Toulouse, France

**Keywords:** platelets, sepsis, endothelium, immunothrombosis

## Abstract

Host defense against infection is based on two crucial mechanisms: the inflammatory response and the activation of coagulation. Platelets are involved in both hemostasis and immune response. These mechanisms work together in a complex and synchronous manner making the contribution of platelets of major importance in sepsis. This is a summary of the pathophysiology of sepsis-induced thrombocytopenia, microvascular consequences, platelet-endothelial cells and platelet–pathogens interactions. The critical role of platelets during sepsis and the therapeutic implications are also reviewed.

## 1. Introduction

The definition of sepsis has recently been modified. According to the Third International Consensus Definitions for Sepsis and Septic Shock, it is defined as a “life-threatening organ dysfunction caused by a dysregulated host response to infection” [1]. It is commonly admitted that sepsis management requires monitoring and intervention, including admission to the emergency department of the intensive care unit, if necessary. When circulatory and cellular metabolism anomalies occur, sepsis is called “septic shock”. These anomalies significantly increase morbimortality. The number of hospitalizations for sepsis continues to grow, which highlights the importance of having a clearer understanding of the pathogenesis to aid in future improvements [2].

The development of multiple organ failure (MOF) increases sepsis-related mortality. MOF is partly due to endothelial dysfunction with hyperpermeability and microvascular thrombosis. Extensive microvascular thrombosis impairs oxygen delivery to cells [3]. This phenomenon results in tissue ischemia and cellular hypoxia leading to partial or complete inhibition of organ function. For example, hepatic arterioles occlusion leads to a decrease in hepatic function and hepatic impairment. Platelets and coagulation are both involved in thrombosis which is generally considered to be a pathological deviation of hemostasis [4]. However, recent findings suggest that intravascular thrombosis also involves processes that are distinct from hemostasis and which occur mainly in pathological situations such as sepsis. The participation of neutrophils and monocytes, as well as dendritic cells leads to a “thrombosis-related signature” which initiates and propagates fibrin formation and triggers platelet activation during the development of thrombosis. Recent works mention a phenomenon named “immunothrombosis”. This suggests that under certain circumstances, thrombosis is a physiological process which constitutes an effective mechanism of innate immunity in which platelets play an important part [4].

## 2. Thrombocytopenia is Common in Sepsis and is Correlated to Mortality

Thrombocytopenia is generally defined as platelet count <150 G L^−1^ and it is classified as severe if platelet count is <50 G L^−1^ [5,6,7]. It is the most common hemostatic disorder in the intensive care unit (ICU) with a prevalence of approximately 50% [8]. Interestingly, platelet count is a part of the SOFA score (Sepsis-related Organ Failure Assessment) [9] which aims to assess the severity of organ dysfunction in critically ill patients.

Between 5% and 20% of patients develop severe thrombocytopenia that can be associated with bleeding and even a moderate-degree thrombocytopenia is associated with organ failure and an unfavorable prognosis. Thrombocytopenia in patients admitted to the ICU is well recognized as a poor prognostic sign [9,10,11] and is associated with a prolonged ICU stay [10,12,13]. Akca et al. reported that there was a biphasic temporal pattern in the way platelet counts changed in a large population of medical and surgical ICU patients [14]. They found an initial acute decrease followed by an increase in the platelet count. The authors showed that thrombocytopenia was associated with an increase in the rate of mortality. Moreover, mortality was associated with prolonged thrombocytopenia and the absence of a relative increase in platelet count. Platelet count reached nadir on day four. Thrombocytopenia was more predictive of death on day 14 than earlier in the ICU (66% mortality versus 16% for patients without thrombocytopenia, *p* < 0.05). At the same time in a large database of critically ill medical and surgical patients, Moreau et al. demonstrated that a 30% decline in platelet count on day 4 strongly predicted hospital mortality which is a better predictive value than the absolute number [15]. Vanderschueren et al. reported that the SAPS II (Simplified Acute Physiology score) and occurrence of thrombocytopenia remained significantly related to ICU mortality (odds ratio (OR) = 4.2, 95% confidence interval (CI) 1.8–10.2) [13].

Sepsis accounts for approximately 50% of all thrombocytopenia in the severely ill [16]. In septic patients, thrombocytopenia is frequently associated with a dysregulated host response [17,18]. The SAPS II is an inexpensive and easy-to-perform test that could serve as an early alert for clinicians.

Thrombocytopenia is a risk marker, rather than a cause, of mortality in the ICU. Clinicians confronted with this hemostatic disorder or a significant decrease in platelet count should actively identify and try to correct the underlying cause(s). A detailed history and careful physical examination are keys to achieving the right diagnosis. This should be supported by some laboratory results along with an interpretation of the data within the clinical context.

## 3. Mechanisms that Contribute to Thrombocytopenia in Sepsis

Thrombocytopenia is a common and multifactorial phenomenon occurring during sepsis. The main causes are decreased platelet production, hemodilution, platelets consumption, increased sequestration of platelets in microvessels, and immune-mediated destruction of platelets. The combination of a decrease in the production associated to an increase of platelets consumption and destruction coexists.

The first step for the clinician consists of eliminating pseudo-thrombocytopenia due to an artefact of laboratory parameters cause by in vitro platelet agglutination in EDTA-anticoagulated blood. Measuring normal platelet count in citrated blood is usually sufficient to confirm the diagnosis.

During sepsis, peripheral mechanisms of thrombocytopenia are preponderant, and a myelogram is therefore unnecessary, except in specific cases. Classically, “platelets consumption” via thrombin-mediated platelet activation is the most common mechanism. In severe forms of sepsis, disseminated intravascular coagulation (DIC) can occur and is characterized by the widespread activation of coagulation, which results in the intravascular formation of fibrin and ultimately thrombotic occlusion of small and midsized vessels [19]. It is an acquired disorder that occurs in a wide variety of clinical conditions, and particularly in septicemia. The activation of diffuse coagulation is triggered by cell-specific membrane components of the microorganism, such as endotoxin, lipopolysaccharide or bacterial exotoxins. The use and subsequent depletion of platelets and coagulation factors at the same time, resulting from the ongoing coagulation, may induce severe bleeding.

Another diagnosis that should not be overlooked in sepsis thrombocytopenia is acquired hemophagocytic lymphohistiocytosis (HLH) [20]. This rare pathology is characterized by clinical and biological abnormalities resulting from the dysregulated activation and proliferation of lymphocytes, leading to an overproduction of cytokines. The main clinical and biological data consists of fever, spleen and liver enlargement, cytopenias (thrombocytopenia, anemia and leukopenia), liver dysfunction, high serum levels of triglycerides and ferritin, and histological evidence of hemophagocytosis. Thrombocytopenia is often a warning signal when associated with other criteria suggested by the Histiocyte Society [21]. Bone marrow evaluation may reveal macrophage activation with hemophagocytosis. In severe HLH cases, multi-organ dysfunction develops and eventually leads to death, demonstrating the importance of a prompt diagnosis for early initiation of treatment.

Activated platelets have also been shown to promote neutrophil recruitment to the site of injury [22] and the formation of neutrophil extracellular traps (NETs) which trap and help kill pathogens [23]. Recently, extracellular histones were described as a cause of thrombocytopenia in critically ill patients [24]. Fuchs et al. demonstrated that NETs, which are extracellular DNA fibers comprising histones and neutrophil antimicrobial proteins, were formed inside the vasculature in infectious and noninfectious diseases [25]. They reported that NETs provided a heretofore unrecognized scaffold and stimulus for thrombus formation. NETs perfused with blood cause platelet adhesion, activation and aggregation, whereas NETs formation in animal models caused rapid and profound thrombocytopenia.

Platelet aggregation/adhesion to leukocytes and endothelial cells is a common mechanism for a type of thrombocytopenia called “immune thrombocytopenia”. Platelet-associated IgG (PAIgG) are found in 30%–40% of septic patients [12,26] and, in the ICU, 30% of thrombocytopenic patients are positive for PAIgG. This population is described as being more subject to sepsis and to have a medical history of cardiopulmonary bypass [27].

In summary, the mechanisms for thrombocytopenia in sepsis are multiple. But in most cases, sepsis resolution allows the slow resolution of this hemostatic disorder. Table 1 summarizes the main etiologies.

## 4. The Role of Platelets in Sepsis

Thrombocytes play a complex role in sepsis as they are able to modulate not only their own function but also that surrounding of cells. During sepsis, coagulation cascades and inflammatory response, together with endothelial tissue damage, constantly cause the activation of platelets which can be further stimulated by direct interactions with pathogens (Figure 1) [28]. During hemostasis, platelets adhere and aggregate at sites of endothelial injury to form a plug which warrants vascular integrity and prevents hemorrhage. In fact, when the vascular wall is damaged, platelets immediately adhere to the subendothelium, newly exposed via the vWF (von Willebrand factor), various collagens, fibronectin, fibrinogen and other adhesive molecules such as laminin and thrombospondin. The adhesion of platelets to the injured subendothelium is ensured by three types of receptors: the Glycoprotein GPIb-V-IX glycoprotein complex (vWF receptor), Glycoprotein VI (GPVI) and α_2_β_1_ integrin (collagen receptors), and α_IIb_β_3_ integrin. At first, the circulating platelets interact with the vWF linked to collagen fibers via the GPIb-V-IX complex and subsequently with collagen via the α_2_β_1_ integrin and the GPVI glycoprotein.

These interactions allow transient adhesion of platelets to the surface of the exposed subendothelium. Platelets can then either detach from the subendothelium and return to the bloodstream (i.e., undergo translocation as long as the disc shape is maintained), or undergo rotation or "rolling" with a change in the shape of the pads that then become spherical. Platelets activate and secrete the contents of their granules. This activation also leads to a conformational change in the α_2_β_1_ and α_IIb_β_3_ integrins which can then only bind respectively to collagen and fibrinogen. These two integrins allow stable and firm adhesion of platelets to the subendothelium, which is then able to spread out and form a platelet monolayer.

During the various processes described above, platelet activation is amplified by soluble agonists secreted or generated by the platelet, such as ADP (Adenosine diphosphate) or TXA_2_ (thromboxane A_2_), or from the coagulation cascade, such as thrombin. Thrombin is the most effective activator of platelets capable of inducing a change in their shape, secretion, and aggregation. It is also the main effector in coagulation, allowing the transformation of fibrinogen to fibrin for thrombus consolidation [29]. TXA2 is a prostanoid produced from arachidonic acid by the action of COX-1 (Cyclooxygenase-1) and thromboxane synthase. As a result of its short half-life, the action of TXA2 is highly localized. Dense granules contain a high concentration of ADP which is salted out during platelet activation. Although considered a weak platelet agonist, it has now been recognized that it plays a role, in vivo, in multiple stages of thrombosis. On activation, ADP is secreted at the site of vascular injury or it amplifies the platelet response and contributes to the stabilization of the thrombus. The interaction of soluble agonists (ADP, thrombin and TXA2) with their seven heterotrimeric G-protein-coupled transmembrane domain receptors generates "inside-out" signaling, leading to the activation of αbβ3 integrin. This mechanism enables the recruitment of circulating platelets in the thrombus which also affects its growth. Furthermore, it is becoming clear that platelets are involved in other processes such as immunity.

To begin with, platelets are able to induce the acute phase response to infection [30,31]. This acute phase response corresponds to the production of proteins such as complement proteins, fibrinogen, and C-reactive protein. These proteins destroy or inhibit the growth of microorganisms and exert procoagulant effects that may limit infection by trapping pathogens within local blood clots. During activation of the acute phase response, platelets are known to be a major source of interleukin-1β (IL-1β) which is not granule stored but is produced upon platelet stimulation (after splicing of pre-messenger RNA (mRNA)). Platelet-derived IL-1β plays a major role in inducing this acute phase response to infection.

In addition, platelets promote innate immune cells responses. Neutrophils and monocytes are the first line of innate immune defense against infection. Activated platelets drive responses in target leukocytes that modulate host response to infection. After platelet activation, they express P-selectin on their surface. Platelet–leukocyte interactions engage the P-selectin receptor P-selectin glycoprotein ligand-1 (PSGL-1) on neutrophils and monocytes [32]. Zarbock et al. showed that platelet–leukocyte interaction via P-selectin was a crucial step in the activation and recruitment of leukocytes to the lung in acute lung injury (ALI) [33].

Interactions between platelets and neutrophils are critical for cell trafficking and to deliver molecular signals. In fact, the formation of neutrophil extracellular traps (NETs) is a prime example. Clark et al. demonstrated that platelet toll-like receptor 4 (TLR4) detected TLR4 ligands in blood and induced platelet binding to adherent neutrophils [34]. These interactions lead to robust neutrophil activation and the formation of NETs. NETs ensnare bacteria within the vasculature, primarily in pulmonary capillaries and liver sinusoids. These NETs have a proteolytic activity that can trap and kill microbes in tissues. Data from Clark et al. suggest that this event only happens under extreme conditions such as severe sepsis. It should also be noted that NETs formation has recently been reported in various diseases such as non-autoimmune and auto-immune disorders [35,36,37]. In fact, in addition to their role in the host defense, recent data suggest that the formation of NETs contributes to the pathophysiology of many diseases such as diabetes, auto immune and renal diseases and heparin-induced thrombocytopenia [38,39,40,41,42]. Sreeramkumar et al. reported that neutrophil recruited to injured vessels in an animal model of inflammation, extended a domain into the lumen, where PSGL-1 clusters scanned for the presence of activated platelets [43]. Their findings revealed that the dynamic reorganization of neutrophil domains and receptors allow simultaneous interactions with both the vascular wall and activated platelets in circulation, to provide a rapid and efficient regulatory mechanism in the early inflammation process. These observations underscore the crucial role of platelet–leukocyte interactions which participate in the defense against infections during inflammation and sepsis.

Toll-like receptors are known to promote NETs formation. They are a highly preserved family of pattern recognition receptors (PRR) that bind pathogen-associated molecular patterns (PAMPs) molecules that are broadly expressed by many infectious organisms. One of the most studied PAMPs is lipopolysaccharide (LPS), which is a part of the gram-negative bacteria membrane, and it is a major TLR ligand. Platelets express numerous TLR family members. Platelet TLR4 signaling leads to platelet activation, the shedding of IL-1β-rich microparticles, and platelet interactions with others cells [44]. TLR2 platelet stimulation increases P-selectin surface expression, the activation of the integrin *α*IIb*β*3, the generation of reactive oxygen species, and in human whole blood, the formation of platelet–neutrophil heterotypic aggregates [45]. Other TLRs may also have platelet-related functions but their role in inflammation and sepsis remains to be studied.

Platelet adhesion and activation, vascular endothelial cell activation, innate immune cell recruitment, NET formation and fibrin deposition, all contribute to an increased propensity of thrombosis in septic conditions [46,47]. Thrombosis and microthrombosis play a major role in innate immunity and “immunothrombosis”, which is the term that describes this process [4]. Immunothrombosis constitutes a line of host defense that is supported by several specific molecular mechanisms that fight against pathogen dissemination and survival. In fact, innate immune cells (especially monocytes and neutrophils) trigger immunothrombosis by their local accumulation in microvessels. They generate a procoagulant surface on microvascular endothelial cells with local delivery of tissue factor, degradation of endogenous anticoagulants, and the provision of a procoagulant matrix consisting of extracellular nucleosomes. Consequently the recruitment of platelets leads to clot growth and NETs formation by neutrophils, resulting in bacterial entrapment in microvasculature [48]. The procoagulant action of NETs involves the activation of factor XII [49] and the clearance of anticoagulants such as tissue factor pathway inhibitor (TFPI) and probably thrombomodulin [50]. In the immunothrombosis phenomenon, fibrin plays a crucial role since it has direct antimicrobial activity and may limit the spread of pathogens [51]. Indeed, fibrin network helps by retaining microorganisms circulating in the blood. Fibrin formation can be initiated by extracellular nucleosomes from NETs resulting in the direct activation of the coagulation contact pathway. Factor XII is thus activated in factor XIIa. At the same time, histones released into the NETs can result in platelet activation via toll-like 2 (TLR2) and TLR4 receptors [47,52,53]. Fuchs et al. were the first to demonstrate that NETs were able to induce platelet-adhesion molecule deposition and thrombin-dependent fibrinogen conversion into fibrin [25].

On the other hand, platelets also influence acquired immune responses including T-cell functions. They may activate platelets through an interaction involving T-cell CD40L (CD40 ligand) and platelet CD40 causing them to release RANTES (Regulated upon Activation, Normal T cell Expressed, and Secreted) which binds to endothelial cells and mediates T-cell recruitment [54]. Platelet CD40L is able to interact with many other cells besides lymphocytes and dendritic cells, granulocytes, fibroblasts, macrophages and monocytes, including other platelets [55,56,57].

## 5. Platelets and Endothelial Cells Interactions During Sepsis

Endothelial cells play a central role in a host’s response to sepsis, including inflammation, coagulation, and vascular permeability. Endothelial cells are not classically considered to be immune cells but they express innate immune receptors such as TLRs, which can be activated by pathogen components or a myriad of host-derived factors, including complement, cytokines, leucocytes, fibrin, and activated platelets and leukocytes [58].

The surface of the vascular endothelium is covered by a structure called a “glycocalyx”, a gel-like layer, which regulates thrombus formation, vascular permeability, and inflammation. The glycocalyx is comprised of a membrane-binding domain containing core proteins (such as proteoglycans and glycosaminoglycans) and plasma proteins (such as albumin and antithrombin). Under inflammatory conditions, the glycocalyx is disrupted by glucuronidases, reactive oxygen species (ROS), and other proteases. The shedding of this structure results in the synthesis and exposure of adhesion molecules, such as P-selectin, E-selectin and Intercellular Adhesion Molecule 1 (ICAM-1), and subsequently in the recruitment of leukocytes and platelets [59,60]. On the other hand, the activated endothelial TLRs induce a shift to an endothelial procoagulant phenotype, with a decrease in the synthesis of tissue factor pathway inhibitor (TFPI), tissue plasminogen activator (tPA) and heparan, and an increased expression of tissue factor (TF) and plasminogen activator inhibitor 1 (PAI-1) [58]. TF exposure to blood leads to thrombin formation which in turn activates platelets and converts fibrinogen into fibrin. The activated platelets may in turn also accelerate fibrin production [61].

This complex interaction between circulating platelets, endothelial cells, and subendothelial structures is mediated by cellular receptors on the surface of platelets and endothelial cells, such as integrins and selectins, as described above, and by adhesive proteins, such as fibrinogen and von Willebrand factor (vWF). vWF allows initial platelet adhesion to the injured vessel wall by binding the platelet receptor GPIb-IX-V. Large and ultra-large vWF multimers, the most active forms in the promotion of platelet aggregation are cleaved by A Disintegrin and Metalloproteinase with Thrombospondin type 1 motif-13 (ADAMTS-13). The release of significant ultra-large vWF multimers following endothelial damage, associated with a decrease in its cleavage by a consumption of ADAMTS-13, could explain the increase of the platelet-vessel wall interaction during sepsis [62].

## 6. The Interaction of Bacterial Pathogens with Platelets

Several research teams have evaluated the ability of bacteria to bind to platelets. Interactions are broadly represented by three types of mechanisms: 1) secretion of bacterial products (i.e., toxins that interact with platelets), 2) direct bacterial binding to a platelet receptor, and 3) binding of a plasma protein, which is a ligand for a platelet receptor, to bacteria [63].

Bacteria can induce platelet aggregation with a characteristic lag-time that varies from seconds to several minutes. This aggregation is an “all-or-nothing” phenomenon.

Bacteria secrete toxins that can activate platelets. For example, *Staphyloccocus aureus* secretes a major cytolysin called α-toxin. Arvand et al. showed that α-toxin promotes blood coagulation via the exocytotic release of factor V from α-granules and the enhanced capacity of platelets to bind external factor V leading to the assembly of prothrombinase complexes. In addition, α-toxin binds to the platelet lipid bilayer creating a transmembrane pore that leads to an influx of calcium-triggering platelet activation [64]. Lourbakos et al. showed that cysteine proteases produced by *Porphyromonas gingivalis* (called gingipains) induce an increase in intracellular calcium in human platelets and cause platelet aggregation with an efficiency comparable to thrombin [65]. Gingipains appeared to activate the protease-activated receptors (PAR-1 and PAR-4) expressed on the surface of the platelets.

Platelets can also interact with bacteria via direct interactions. Different species of bacteria contain mimetic ligand motifs that act as agonists on platelet receptors. Lipopolysaccharide from *Escherichia coli* can bind to TLR4 and *S. pneumoniae* lipopeptide to TLR2. Recently, reports have demonstrated that *S. epidermidis*, *S. aureus,* and *S. gordonii* expressed proteins could directly bind to GPIIb-IIIa in the absence of a bridging molecule [66,67,68]. Other platelet surface proteins can bind to bacteria. Bacterial components are able to bind indirectly to platelet receptors via different proteins such as fibrinogen, fibronectin, and von Willebrand factor [69]. In fact, several strains of bacteria can bind to plasma proteins that bridge with their specific platelet receptors, resulting in their activation [69]. For example, *Staphylococcus aureus* can bind vWf that interacts with GPIbα platelet receptor thus triggering its activation [70]. *S. aureus* can also bind via clumping factor A (ClfA) several plasma proteins such as fibrinogen or fibronectin that can in turn bind to GPIIb-IIIa [71,72].

## 7. Therapeutic Implications

Platelet activation contributes to microvascular thrombosis and organ failure in systemic inflammation, particularly in septic conditions. Based on this observation, the question of whether drugs that inhibit platelet activation may have a benefit in critically ill patients arises, particularly in septic shock to limit microvascular occlusion and multiple organ failure.

Preclinical evidence for aspirin and P2Y_12_ inhibitors in sepsis in animal models is already published. In fact, Winning et al. pre-treated mice with clopidogrel for four days prior to intraperitoneal administration of endotoxin (LPS) from *E. coli*. Clopidogrel eliminated the LPS-induced drop in platelet count and reduced fibrin deposition in lung tissue [73].

In a model of abdominal sepsis, Rahman et al. showed that pulmonary infiltration of neutrophils was reduced by 50% in ticagrelor-treated animals [74]. Moreover, ticagrelor abolished CLP-provoked lung edema and decreased lung damage score by 41%. Another study showed that mice injected with *Salmonella enteritidis* endotoxin and pre-treated with aspirin 30 minutes prior to infection exhibit a significant 24-hour survival rate benefit with different dosages of aspirin [75]. However, the results of the animal studies are sometimes contradictory. In a recent article, in a mouse model of septic shock by cecal ligation and puncture, the authors did not find any protective role of the P2Y_12_ purinergic receptor using mice deficient for the P2Y_1_ receptor and with clopidogrel.

The P2Y_12_ receptor has also been studied for his modulation in inflammation. In P2Y_12_ null mice, clopidogrel is shown to have pleiotropic effects, especially on neutrophils, with an effect on their number, even in the absence of a platelet receptor [76].

Class I phosphoinositide 3-kinase β (PI3Kβ) could be an interesting target for antithrombotic therapy in sepsis. In fact, this PI 3-kinase isoform is shown to play an important role in thrombus formation and stability [77,78,79,80,81]. In vivo, isoform-selective PI3Kβ inhibitors eliminate occlusive thrombus formation but do not prolong bleeding time. However, such treatments have not yet been tested in this indication in phase II trials. Interestingly, even though invalidation of the catalytic subunit of PI3K β (p110β) specifically in platelets is known to induce instability of the thrombus at high shear rate [80], it was shown in a mouse model of sepsis that septic conditions reversed the thrombus instability at a high shear rate, bringing out an alternative mechanism enabling platelets to form stable thrombus [82].

Clinical data on the use of antiplatelet agents in critically ill septic patients also reported encouraging results [83]. In a retrospective study, Eisen et al. showed a strong association between acetyl salicylic acid (ASA) and survival in the ICU [84]. The ASA group had a 10.9% in-hospital mortality compared with 17.2% in the nonusers after propensity matching.

However, there is a lack of prospective randomized controlled studies. Valerio-Rojas et al. performed a retrospective cohort study of severe sepsis and septic shock in adult patients. They showed that antiplatelet therapy was associated with a decreased incidence of acute lung injury and acute respiratory distress syndrome.

Otto et al. analyzed the medical records of 886 septic patients who were admitted to the surgical ICU of a university hospital [85]. Logistic regression analysis indicated that patients who were treated with ASA (100mg/d) had a significantly lower mortality.

Given the inherent limitations of observational studies, only randomized controlled trials could answer the question concerning the interest and effectiveness of antiplatelet agents to reduce organ failure and morbimortality. Currently, there are two ongoing relevant clinical trials. “Aspirin for the treatment of sepsis” (NCT01784159) involves the beneficial effect of seven days of aspirin treatment on organ dysfunction and the duration of ventilation in severe septic patients. The other trial is “Aspirin to inhibit sepsis (ANTISEPSIS, ACTRN12613000349741) which examines the effect of daily aspirin administration on mortality and admission to the ICU for sepsis. The results of these trials will help to elucidate the role of aspirin in treating sepsis.

## 8. Conclusions

Platelets play key roles against infection and are involved in various mechanisms to promote the immune response and the activation of coagulation. Thrombocytopenia is common in the ICU during sepsis, causes are multiple, and low platelet count is correlated with poor outcome. A better understanding of platelet activation mechanisms and crosstalk between endothelial cells, immune cells, and pathogens would provide the perspective to target several deleterious pathways in sepsis, particularly in platelet activation.

## Figures and Tables

**Figure 1 ijms-20-03494-f001:**
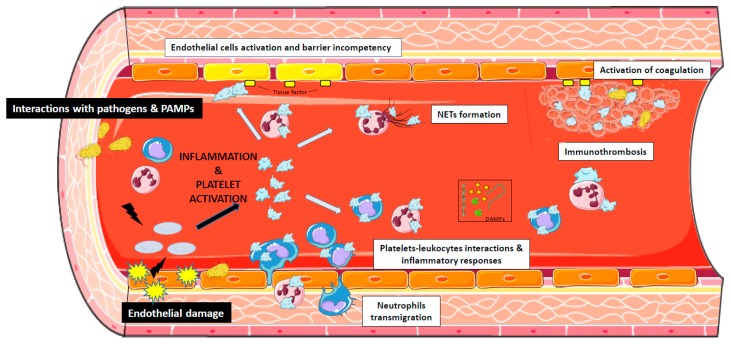
Selected examples of pro-inflammatory role of platelets during sepsis: endothelial damage and interactions with pathogens such as pathogens-associated molecular patterns (PAMPs) activate platelets that interact with endothelium, with monocytes and neutrophils, promoting neutrophils extra-cellular Trap (NET)osis, neutrophils transmigration, activation of coagulation through tissue factor release and immunothrombosis. DAMPs: damage associated molecular patterns (microvesicles, free DNA, protease).

**Table 1 ijms-20-03494-t001:** Etiologies of thrombocytopenia during sepsis.

**Main Etiologies of Thrombocytopenia during Sepsis**
**Pseudothrombocytopenia**
Laboratory artefact (*in vitro* agglutination in EDTA-anticoagulated blood)
**Decreased platelet production**
Viral infection (EBV, CMV, HCV, HIV)
Bone marrow suppression due to medication (antibiotics, proton pump inhibitor)
**Hemodilution**
Massive vascular infusion of fluids
**Increased platelet consumption/sequestration**
Thrombin-mediated platelet activation
Disseminated Intravascular Coagulation
Acquired hemophagocytic lymphohistiocytosis (HLH)
Platelet aggregation/adhesion to leukocytes and endothelial cells
Thrombus formation in extracellular DNA fibers from neutrophil extracellular traps (NETs)
**Immune-mediated destruction**
IgG antibodies associated to platelets (PAIgG)
Autoantibodies directed against platelets glycoproteins
Heparin-induced thrombocytopenia

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
