# Peer review of "Platelets Are Critical Key Players in Sepsis"

_ijms, 2019, doi:10.3390/ijms20143494_

Round 1
Reviewer 1 Report
This short review provides an update on platelets in the pathogenesis of sepsis. The review proceeds in a logical order and covers many of the key issues. Close attention needs to be paid to grammar, spelling and other typographical errors. There are no major issues, but the manuscript could be improved in a number of areas.
1. A section on the basic biology of platelets in haemostasis defining the function of key platelet receptors mentioned in the review and the molecular mechanism of how platelets facilitate coagulation and interact with leucocytes and endothelial cells would be helpful for the naive reader less familiar with the platelet field.
2. The text in lines 159 and 160 implies that NET formation only occurs in sepsis but NETosis occurs in other diseases such as in patients with type 2 diabetes. The text should be modified accordingly.
3. Line 241 should read "Platelets can also interact with bacteria via direct interactions."
4. Section 7 on Therapeutic Implications needs elaboration. It does not address the issue of potential hemorrhage if anti-platelets are used in thrombocytopenic patients with or without associated coagulopathies. It should clarify that aspirin and P2Y12 antagonists are not platelet specific and may be of benefit due to non-platelet effects. For example, the P2Y12 receptor is expressed on endothelial cells. In addition, a key paper questioning whether P2Y12 receptor antagonists are of benefit in sepsis is not cited
A Potential Protective Role of Platelets during Septic Shock Does Not Depend on Their Purinergic Receptors Beatrice Hechler, Chloé Zimmermann, Yannick Rabouel, Stéphanie Magnenat, Mélanie Burban, Julie Boisramé-Helms, Ferhat Meziani and Christian Gachet
Blood 2016 128:2537;
Further some of the effects of clopidogel in models of sepsis appear to be due to off-target effects.
P2Y12 Receptor Modulates Sepsis-Induced Inflammation Elisabetta Liverani, Mario C. Rico, Alexander Y. Tsygankov, Laurie E. Kilpatrick, and Satya P. Kunapulia,Arterioscler Thromb Vasc Biol. 2016 36: 961–971.
Finally, it should be clarified that PI3Kbeta inhibitors while of interest amongst a wide range of new potential antiplatelets have not yet proceeded past phase II trials and are therefore not yet clinically available.
Reviewer 2 Report
The review "Platelets are critical key players in sepsis" provides an overview about the current literature con this topic. Platelets are recently being recognised as immune players in many infectious and inflammatory pathologies; so that, this review is relevant in this field. The text is sound and readable; however, there are few comments that I think may help understanding.
- I first have one curiosity regarding sentence in line 40: are immune cells responsible for fibrin formation? Or is fibrin formation causing immune cell activation and participation?
- There is repetitive information in lines 66-68 and 77-81. Please, try to rephrase avoiding repetition.
- Lines 95-120: I find that the information given here is a bit disconnected. I believe that Table 1 summarises in a much clearer way the information revised. I know it is a revision of mechanisms for thombocytopenia in sepsis, but I think it is worth trying to rephrase these paragraphs to help the reader understanding.
- Line 228-229: In the same way, I think mechanism 3 is not clear enough.
